# A Scoping Review of Digital Well-Being in Early Childhood: Definitions, Measurements, Contributors, and Interventions

**DOI:** 10.3390/ijerph20043510

**Published:** 2023-02-16

**Authors:** Simin Cao, Hui Li

**Affiliations:** 1Shanghai Institute of Early Childhood Education, Shanghai Normal University, Shanghai 200234, China; 2Macquarie School of Education, Macquarie University, Sydney, NSW 2109, Australia

**Keywords:** young children, early childhood, digital, well-being, review, definition, measurement, contributor, intervention

## Abstract

Digital well-being concerns the balanced and healthy use of digital technology, and the existing studies in this area have focused on adolescents and adults. However, young children are more vulnerable to digital overuse and addiction than adults; thus, their digital well-being deserves empirical exploration. In this scoping review, we synthesized and evaluated 35 collected studies on young children’s digital use and their associated well-being that were published up to October of 2022 to understand the related definitions, measurements, contributors, and interventions. The synthesis of the evidence revealed that (1) there was no consensus about the definition of the concept of digital well-being; (2) there were no effective ways of measuring young children’s digital well-being; (3) both child factors (the duration and place of digital use, as well as the child’s demographic characteristics) and parent factors (digital use, parental perception, and mediation) contribute to young children’s well-being; and (4) there were some effective applications and digital interventions reported in the reviewed studies. This review contributes to the development of this concept by mapping the existing research on young children’s digital well-being, as well as proposing a model and identifying the research gaps for future studies.

## 1. Introduction

The COVID-19 pandemic has caused repeated lockdowns and associated closures of schools and pre-schools that have forced children (under age 18) to learn from home using digital devices [1]. In particular, young children (those aged from 0 to 8 years in this study) are sensitive and vulnerable to digital overuse and problematic use [2]. Therefore, their digital well-being is in crisis or at least compromised. However, the existing studies have focused on the digital well-being of adolescents and adults, leaving the digital well-being of young children unexplored. In fact, young children have started to become ‘digital natives’ since the turn of this millennium [3], and the COVID-19 pandemic has hugely accelerated the digitalization of their lives [1,4]. This drastic transformation has significantly increased digital use and screen time in young children globally [5,6]; thus, many countries have issued national guidelines to tackle the problem [7,8]. These guidelines, however, only provide advice on how to limit young children’s screen time and digital use, which is highly relevant but which is not necessarily translated into improvements in young children’s digital well-being [9]. Moreover, there is a lack of national guidelines specifically related to young children’s digital well-being, which is an emerging and underexplored research area. Therefore, a scoping review is needed urgently in order to map the literature on this topic systematically. To meet this aim, we conducted a scoping review to identify the definitions, measurements, contributors, and interventions that could inform practice and research in this field.

### 1.1. Digital Use and Digital Well-Being

Since the turn of this millennium, our work, social, and leisure environments have become suffused with digital technologies, such as laptops, tablets, and smartphones [10]. These digital technologies connect us to content, contacts, and services without time or place constraints [11]. However, this phenomenon of being “permanently connected” [12] has not only resulted in convenience but has also introduced struggles. Digital use is found to distract users from work and study [13], cause sleep and health problems [14], interfere with social activities [15], lead to procrastination [16] and negative emotions [17], and cause digital addiction [18]. In particular, concerns about the negative impact of digital use have been raised since COVID-19 caused lockdowns, which led to a dramatic increase in people’s digital use time [19]. To cope with these struggles, the topic of digital health and well-being has emerged as a public concern and research topic, with researchers focusing on how to maintain a good and healthy lifestyle in the digital era [20].

Young children, who are increasingly gaining access to digital technologies, are vulnerable to the associated ‘unanticipated risk’ that may challenge their maturity [21]. Accordingly, in 2012, Nansen coined the term ‘digital well-being’ to situate online risk within the concept of well-being, aiming to highlight children’s online resilience and critical capacities to cope with online risks and achieve online safety [22]. Later, Beetham explicitly defined this term for educational purposes: digital well-being refers to the balance between the potential risks and benefits of digital engagement [23]. This definition underscored the supporting role of teachers in students’ digital well-being. Therefore, digital well-being has been widely regarded as a strategy or solution to prevent digital overuse or even digital addiction and it has attracted drastically increased attention during the past decade [2]. Recently, Abeele defined digital well-being as ‘a subjective individual experience’ of optimal balance and health in the use of digital technologies [24]. The above-mentioned definitions, however, are general definitions of digital well-being, and are not specific to young children. On the other hand, Johnston defined digital well-being as how digital use could be integrated into children’s lives to enhance their learning, development, and long-term outcomes [25]. In this scoping review, we thus aimed to collect and analyze all the existing definitions to identify the best-fitting definition of young children’s digital well-being.

Recently, Yue et al. proposed a three-dimensional framework: (1) crafting and maintaining a ‘healthy relationship’ with technology that can be used in a balanced and civil way; (2) identifying and understanding the ‘positive and negative impacts’ of engaging with digital activities; (3) being aware of ways to ‘manage and control factors’ that contribute to digital well-being [26]. This three-dimensional framework, however, has not been verified by other studies. In addition, very few studies have proposed and validated other frameworks of digital well-being. Therefore, in this scoping review we also aimed to explore the existing studies to identify possible frameworks of digital well-being in the early years.

### 1.2. Digital Well-Being in Early Childhood

Evidence demonstrates that early digital use is a double-edged sword for young children. On the one hand, young children can use digital technology for learning, communicating, creating, and entertainment [27,28], which are beneficial for early learning and development [29]. Hence, digital technologies have been labeled ‘early childhood educators’ and ‘integral learning tools’ [30] and accordingly many countries have launched standards or policies to promote children’s digital literacy and digital citizenship [29]. However, on the other hand, inappropriate early digital use may cause severe physical and mental health problems in young children [20], such as obesity [31], sleep disturbance [32], postural effects, and visual disorders [33], as well as potentially hindering cognitive and brain development [34], executive functions [35], and social and emotional development [36]. In particular, Meng et al. (2022) reported that the global rates of smartphone and social media addiction were 26.99% and 17.42%, respectively, and the rates for the South-East Asia region were 41.63% (smartphone addiction) and 59.36% (social media addiction) [19]. Furthermore, Anitha et al. (2021) investigated 613 children aged between 18 months and 12 years and found that the prevalence of screen addiction was 28.1% [37]. Therefore, digital well-being in early childhood has recently emerged as a global concern.

To address this global concern, many national guidelines for young children’s digital usage have been released. For example, the American Psychological Association (APA) has published specific guidelines for children’s digital usage, *Digital Guidelines: Promoting Healthy Technology Use for Children* [8]. These guidelines suggest that children younger than 18 months should not use any screen-based digital technologies except video chatting, and those aged 2 to 5 should not watch screens for more than 1 h per day. These guidelines privilege human communication over other types of communication or content. In addition, according to these guidelines, parents should be responsible for (1) choosing high-quality programming; (2) teaching children about technology from a young age; (3) discussing with children the benefits and risks of technology; (4) establishing family rules for technology use; (5) restricting digital usage to at least 30 min before bed; and (6) helping children to develop social skills through establishing real-life relationships [8]. Subsequently, the Canadian Paediatric Society (CPS) issued the ‘4 Ms’ guide for parents of young children (0–5 years old): ‘minimize, mitigate, be mindful, model’ [38]. Meanwhile, the Department of Health of the Australian government published the *24-h Movement Guidelines for the Early Years*, which recommended more physically active time and less sedentary screen time [39]. Recently, the Indian Academy of Pediatrics (IAP) published the *Indian Academy of Pediatrics Guidelines on Screen Time and Digital Wellness in Infants, Children and Adolescents*, suggesting controlling screen time, maintaining normal routine activities, building a good home environment, and monitoring children’s daily digital usage [7]. The main purpose of these guidelines is to reduce or limit screen time, reflecting negative views and reservations about early digital use [40].

In contrast, in 2021, the US state of Minnesota issued a “Digital Well-being Bill” and provided USD 1 million in funding support for the training of educators, parents, and adolescents, providing them with knowledge and skills to promote physical and mental well-being [41]. This initiative was based on a positive view of early digital usage and aimed to advocate balanced, intentional, and responsible digital usage. However, this meaningful program was designed for school children [21] rather than young children, whose digital well-being remains underexplored [25]. Therefore, a synthesis of the existing evidence related to early childhood digital well-being is urgently needed for policymaking and practical improvement. To address this research gap, in this study we aimed to collect all the studies that have been conducted on digital well-being in young children and provide a knowledge base for policy improvements and future studies.

### 1.3. Research Objectives and Questions

The above literature review demonstrated the inconsistencies and arguments regarding the definitions, nature, and constructs of digital well-being in the early years. Furthermore, we identified the lack of a systematic knowledge framework of young children’s digital well-being across studies. Therefore, in this scoping review we aimed to synthesize, display, and evaluate the latest literature on young children’s digital well-being, focusing on the related definitions, measurements, contributors, and interventions. Based on the previously published literature, in the present scoping review we focused on young children’s digital well-being and aimed further to establish a theoretical basis for exploring the topic in the future. In particular, the following questions guided this study:What is the digital well-being of young children?What are the effective ways of measuring young children’s digital well-being?What are the contributors to young children’s digital well-being?How can young children’s digital well-being be improved?

## 2. Materials and Methods

In this study, we aimed to review the current literature related to young children’s digital well-being (in children under 8 years old). A scoping review approach is useful for clarifying the definition and influence factors in an emerging area [42]. In this scoping review, we followed the guidelines of the *PRISMA extension for scoping reviews* (PRISMA-ScR) statement [43].

### 2.1. Search Strategy

The electronic databases used for the literature search included Scopus, EBSCO, Web of Science, ProQuest, PubMed, and PsycINFO. After discussions among the research team, the following search terms were used in combination: (“well-being” OR “well-being” OR “wellness”) AND (“digital” OR “technology” OR “mobile” OR “smart*” OR “internet” OR “screen”) AND (“young child*” OR “preschool*” OR “kindergarten*” OR “infant” OR “toddler” OR “pre-k*” OR “early childhood”). To explore the scope of digital well-being, we investigated all kinds of academic articles (including peer-review articles, book chapters, dissertations, and technical reports) published up to 26 October 2022.

### 2.2. Inclusion and Exclusion Criteria

For this review, we established three inclusion criteria. First, the articles had to be related to the definition, measurement, influencing factors, or interventions in regard o young children’s digital well-being. Second, the articles had to be full-text studies written in English. Third, the articles initially had to focus on young children (under 8 years old). As few studies focused on young children’s digital well-being, the third criterion was changed to allow studies of children under 18 years old. Meanwhile, articles were be removed based on the following exclusion criteria: (1) articles that focused only on young children’s physical health or medical studies; (2) studies of which the participants or target samples were children with special needs; (3) studies on the design and testing of applications; (4) studies on the well-being of other stakeholders (teachers or parents).

### 2.3. Study Selection

Figure 1 displays a flowchart of the study selection process. As shown in Figure 1, a total of 1677 articles were found by searching the electronic databases. Among these articles, 871 were from Scopus, 85 from EBSCO, 406 from Web of Science, 16 from ProQuest, 271 from PubMed, and 28 from PsycINFO. These articles were filtered in the following steps. First, we removed the duplicate articles (*n* = 913) using Endnote. Second, 764 articles were imported into Covidence and screened by title and abstract, and 653 studies were excluded because of irrelevance, leaving 111 articles for further scrutiny. Third, the 111 articles were assessed for full-text eligibility, and 80 were excluded because they (1) were unrelated to young children (*n* = 47); (2) were unrelated to digital usage (*n* = 11); (3) used a pediatric or adult population only (*n* = 16); (4) had no full-text (*n* = 4); or (5) were not in English (*n* = 2). Finally, after checking the references of the 31 included articles, we found four additional articles and included them in this study. Eventually, 35 articles were chosen and analyzed in this study.

### 2.4. Analysis

The data from 35 collected articles were converted into useful codes, themes, and categories using the “RADaR” Technique [44]. The “RADaR” Technique (Rapid and Rigorous Qualitative Data Analysis) involves the use of spreadsheets and word processing software to develop all-inclusive data tables through several revisions in a ‘rapid and rigorous’ manner. It includes 5 steps: Formatting the data, placing the formatted data transcripts into an all-inclusive table, reducing the all-inclusive data, reducing the data, and obtaining draft project deliverables [44]. In this study, we followed these five steps in the following way.

We summarized the selected articles by author, published years, title, type, method, and main conclusions so that all articles were formatted similarly.We coded and placed the basic information into an all-inclusive table, which included the following row titles: Doc Number, First Author, Publishing year, Title, Type, Method, Theoretical framework, Age of Target Population, Samples, Country, Main Conclusion, Quality, Definition of Digital Well-Being, Measurement of Digital Well-Being, Influencing Factors, Correlation between Digital Usage and Well-Being, Intervention Regarding Digital Well-Being, Research Suggestions, Inclusion Criteria Met (Code 1 or Code 2), Notes, Cite, and DOI.We reduced the amount of useless information. The research team discussed the usefulness of all information in the all-inclusive table and removed useless data. In this phase, the “theoretical framework” column was removed because very few studies reported their theoretical framework. Based on the focused code suggested by Watkins, the information that could not answer the research questions was also removed, such as Cite and DOI.We drafted and coded the theme table. We drafted the themes from the all-inclusive table first and then placed the relevant information into the theme table. Four tables were formulated: the definition table, the measurement table, the contributor table, and the intervention table. Then, each table was coded separately to develop the “focused code” [44].The data used for the analysis included information on authors, titles, keywords, and the main text. We converted all the textual data into useful codes, themes, and categories using the data charting technique [44]. This approach has been widely employed to identify knowledge gaps and research trends. In this study, data charting was performed by the first author independently, with the other author acting as the auditor to ensure the reliability of this study.

## 3. Results

Although it is an important topic, early childhood digital well-being has rarely been studied. Among the 35 studies investigating this topic, only four were conducted in developing countries/regions, including Turkey, India, and Malaysia. The rest were all conducted in developed countries/regions. For example, 11 studies were conducted in the United States and 10 in European countries, including Germany, the UK, Italy, Belgium, Netherlands, Singapore, Italy, Poland, Slovenia, and Spain. In addition, five studies were conducted in Australia and three were conducted in Canada. More characteristics of the included studies, such as the age of the target population, the research method used, and the instruments used, can be found in Table A1.

### 3.1. The Definitions, Nature, and Construction of Digital Well-Being

We analyzed the concept of digital well-being in the literature and encoded it using three dimensions: definitions, nature, and construction.

The definitions reflect our understanding of the nature of the target phenomenon, but there have been arguments about the nature of digital well-being. For instance, Gui et al. defined digital well-being as ‘a state’ in which subjective well-being is maintained in an overabundant digital environment [2], which is, per se, a balance between its adverse effects and its benefits. In contrast, Royal defined it more positively, stating that digital well-being is ‘a way of life’ with digital technology promoting optimal health and well-being. This ‘way of life’ involves the integration of an individual’s body, mind, and spirit to enable them to live more fully within the human, natural, and digital communities. Therefore, it refers to the ideal state of health and well-being that each digital citizen is capable of achieving [45]. Later, Google officially defined the nature of digital well-being as ‘a state of satisfaction’ when digital technology supports people’s intentions [46]. Thus, technology should be designed for ‘crafting and maintaining a healthy relationship with technology’, and the focus should be on how technology serves us and moves us towards our goals rather than distracting and interrupting us. Hence, Google launched an application named ‘digital well-being’, empowering users to track their usage time on various applications and balance their digital and non-digital activities. The same function could be found in iPhones and iPads in the ‘Screen Time’ application. Since then, digital well-being has been discussed frequently by technology designers, educators, policymakers, and so on. In summary, the existing literature tends to define the nature of digital well-being as ‘a state of balance’ [2], ‘a way of life’ [45], and ‘a state of satisfaction’ [46].

Recently, Abeele defined digital well-being as ‘a subjective individual experience’ of finding an optimal balance between the advantages and disadvantages associated with digital use [24]. These digital well-being experiences, as a dynamic system composed “of affective and cognitive appraisals of the integration of digital connectivity into ordinary life”, are influenced by person-, device- and context-specific factors [24], so that digital well-being can be improved through interventions by “disrupting the system” [24]. Similarly, the JISC stated that digital well-being is one of the elements in the digital capabilities framework, which can be improved through appropriate management and support [47]. The Digital Well-Being Bill passed by the Minnesota legislature in 2021 also claims that digital well-being refers to the “balanced, intentional, and responsible use of technology so that we can thrive and be well”, and that it can be improved through training or education [48]. In brief, it is recognized that digital well-being can be improved through specific approaches, such as management, education, and interventions.

Through a synthesis of the above definitions, we found that the word “balance” was commonly used when defining digital well-being, referring to an equilibrium between maximizing the benefits and minimizing the harm of digital use. Four types of balance were found. (1) Johnston (2021) defined digital well-being as ‘a balance’ between offline and online life [25]; (2) Abeele (2021) defined it as ‘a subjective individual experience’ of optimal balance between the benefits and drawbacks [24]; (3) Yue (2021) defined it as a healthy relationship with technology that can be used in a ‘balanced and civic way’ [26]; and (4) the JISC (2019) defined it as using digital technology in a balanced way, to ‘enjoy the benefits and avoid potential risks of digital use’ [47]. Therefore, the core nature of digital well-being is “balance”. Moreover, some other articles and guidelines have indicated various meanings of “balance” in relation to digital use, for example, the balance between screen time and other activities [25,49], between traditional play and digital play [25], between action (challenge) and the capacity to act (skills) [25], between work and life when managing a digital workload, between overload, and distraction [50], between an individual’s resources and the challenges they face [51], between the use of technology in and outside of the classroom [41], as well as balance in regard to parental guidance/education and regulation [51]. According to the JISC, digital well-being is a complex concept that can be viewed from a variety of perspectives: digital personal well-being and digital social well-being, and the balance between its negative and positive effects on an individual’s well-being can be considered in four contexts: social, personal, learning, and work [47]. From the personal perspective, there are at least two kinds of balance: the “hedonic balance”—the balance of positive and negative emotions within a person [52]—and the “eudaimonia balance”—the balancing of resources and challenges that brings a person a sense of autonomy, mastery, purpose, and connectedness with people, as well as a sense of “flourishing” [52]. Meanwhile, from a social perspective, parents and teachers are asked to balance digital resources and digital usage in the home and school environments [41,51].

Some studies have further explored the constructs of digital well-being. Initially, McMahon and Aiken (2015) proposed a three-part model of digital well-being: physical well-being (e.g., posture), mental well-being (e.g., the level of attachment to devices, impulsiveness in responding to device notifications), and psychosocial well-being (e.g., online security, privacy) [53]. Later, the JISC (2019) in the UK followed this three-part model, referring to physical, mental, and emotional health [47]. Recently, Johnson (2020) proposed three levels of digital well-being: physiological, behavioral, and emotional [54]. These studies jointly indicated that digital well-being can include at least three constructs: physical, mental, and social-emotional. However, Yue (2021) proposed a comprehensive nine-part model of digital well-being: digital safety and security, digital rights and responsibilities, digital health and self-care, digital creativity, digital emotional intelligence, digital communication, digital consumerism, digital employment and entrepreneurship, and digital activism/civic engagement [26]. In addition, Yue added three dimensions to the digital well-being framework to further reflect the relationship between digital well-being and digital citizenship: digital skills, identity, empowerment and agency [26]. This framework, however, has some overlaps with digital citizenship or digital literacy. However, Yue is not alone, as the JISC also placed digital well-being in the digital literacy framework [47], Vissenberg also described digital literacy as similar to digital well-being [50], and Johnston indicated that digital well-being should overlap with digital citizenship [25].

### 3.2. Measurements of Digital Well-Being

Ong (2021) reviewed 63 relevant studies and found no specific online well-being scale [52]. Despite the lack of scales precisely designed to measure an individual’s digital well-being, some scholars have tried to measure young children’s well-being and digital use separately. In the existing studies reporting measurements of young children’s well-being, social and emotional competence was the most commonly measured characteristic. Przybylski (2021) measured 19,930 young American children’s digital screen time and psychological well-being, making measurements based on parental reports of their responses to four questions: caregiver attachment, resilience, curiosity, and positive affect in the past month. However, these variables have not been combined into a composite well-being measure, and the reliability of these items was relatively low (a = 0.57) [55].

Therefore, some other pediatric scales have been borrowed to measure early digital well-being. Monteiro (2021) adopted the Baby Pediatric Symptom Checklist (BPSC) to test the emotional and behavioral problems of infants (younger than 18 months) and the Preschool Pediatric Symptom Checklist (PPSC) to test the emotional and behavioral problems affecting young children (from 18 to 66 months) [56]. Both were developed from the Pediatric Symptom Checklist (PSC). As part of the survey of the well-being of young children, BPSC and PPSC showed strong internal and retest reliability [57]. Oliva (2021) employed BPSC and PPSC to explore the risks and protective factors of the mental health symptoms of Italian children during the COVID-19 pandemic [58].

In addition, traditional parent surveys such the Strengths and Difficulties Questionnaire parent version (SDQ) have also been borrowed to measure the outcomes of young children’s digital usage [59,60,61]. For example, Tezol adopted SDQ to assess the psychosocial well-being of young children (aged from 2 to 6). SDQ is a widely used instrument to measure mental health problems in children and adolescents, as it has good reliability and predictiveness of mental disorders in preadolescence [62].

However, the above measurements focused on ‘well-being’, leaving the other keyword ‘digital’, unmeasured. Recently, Byrne et al. (2021) reviewed different measures of screen time among young children (ages 0–6) and found that 60% of the measures only assessed screen time by inquiring into one to three items, and few (11%) measures assessed the content of media. Furthermore, 24% of articles measured television watching only, whereas only 3% focused on young children’s digital usage (e.g., their usage of smartphones and tablets). Furthermore, the instruments’ psychometric properties (reliability and validity) were rarely reported in these articles. Therefore, Byrne highlighted the need for improved measurement tools to capture the complexity of digital use and digital well-being [63].

Obviously, there is a need to develop a specific scale to measure young children’s digital well-being [64]. Domoff (2019) developed a scale to measure the problematic media usage of children aged 4 to 11 years based on nine criteria for Internet gaming disorders in the DSM-5 [65]. This scale had high internal consistency and validity (Cronbach α = 0.97) [66]. It was composed of 27 items that could be grouped into five dimensions: emotional symptoms, conduct problems, hyperactivity/impulsivity, peer relationship problems, and prosocial behaviors. Based on these 27 items, Domoff selected nine items to develop the Problem Media Use Measurement Scale—Short Form (PMUM-SF) and reported that the Cronbach alpha value reached 0.93, confirming it as a reliable instrument. Accordingly, Domoff’s measurement tool has already been validated and used in Arabian, Chinese, and Spanish societies [67,68,69].

### 3.3. Contributors to Digital Well-Being in Young Children

Many studies have explored the factors associated with young children’s digital well-being, which can be classified into three domains: (1) child variables, including the child’s digital usage (e.g., the duration and place of digital use) and the child’s demographic characteristics; (2) parent variables, including parents’ digital use, parental perception, and mediation; and (3) social and context variables, including family SES, digital content, and country differences.

#### 3.3.1. Child Variables

First, screen time (or the duration of digital use) was found to be the most influential factor in regard to young children’s well-being [63], and in most cases, the impact was negative. For example, Ricci found that excessive screen time resulted in a higher risk of the fear of sleeping alone and fear of the dark [32]. Stiglic and Viner reviewed 13 articles and concluded that higher levels of screen time were associated with various health harms such as obesity, unhealthy diet, depressive symptoms, and lower quality of life [70]. However, some studies reported a positive association between digital usage and young children’s well-being. For example, in a longitudinal study, Hinkley et al. (2017) reported a positive association between sedentary digital usage and young children’s intrapersonal and stress management and social and emotional skills [71]. This discrepancy might be caused by the varying effects of digital use, which might depend not only on screen time but also on the type and content of media accessed and the characteristics of the individual child [72].

Second, the place in which a digital device is used is another influential factor related to young children’s digital usage and well-being. For example, in a cohort study including 907 girls and 952 boys participants, Pagani (2019) found that the children who lived in a bedroom with a television at age 4 had higher body mass index values and more unhealthy eating habits, as well as higher levels of emotional distress, depressive symptoms, victimization, and physical aggression and lowers levels of sociability at the age of 12 or 13 [73].

Third, demographic characteristics such as age, gender, and birth order could affect young children’s digital use and well-being. After an analysis of nearly twenty thousand children aged 2–5 years, Przybylski found that daily digital screen usage increased with age, and that those who used digital technology the most were male, non-White children with less educated caregivers, who lived in less affluent households [55]. Tezol found that the amount of time spent on digital technologies significantly differed according to young children’s age, gender, screen time, birth order, and first screen exposure [61].

#### 3.3.2. Parent Variables

Generally, parents play three key roles in young children’s digital usage, as facilitators, teachers, and gatekeepers [74]. Parents’ digital usage and their perception and mediation of young children’s digital usage jointly influence young children’s digital use and well-being [75]. First, parental digital usage affects young children’s digital usage. Wong et al. (2020) investigated the parents of 1254 three-year-old children in Hong Kong and found that parent distraction and problematic digital technology usage could be used as predictors of their children’s screen time and psychosocial difficulties [59]. Similarly, by investigating 477 parents of kindergarteners (ages 3–6) in China, Li et al. (2022) found that parental screen addiction could affect young children’s screen addiction both directly and indirectly. This effect was mediated by parental anxiety and the parent–child relationship [76].

Second, parents’ perceptions and mediation of early digital usage play a key role in young children’s digital usage and well-being. Parents with a negative view of young children’s digital usage may employ more restrictive strategies to mediate young children’s digital usage. In contrast, parents who perceive early digital usage positively prefer to mediate young children’s digital usage by actively talking or co-viewing [77,78]. These different mediation strategies may cause various outcomes for young children’s well-being. For example, parental guidance and support can result in cognitive or social-emotional benefits and self-regulation in children’s digital engagement [25]. In contrast, restrictive strategies have no such helpful outcomes and are less effective in controlling digital use time [79]. Therefore, by encouraging active discussion and supporting early digital usage, parents could optimize the messages provided by positive digital content [80] and build children’s skills and agency [25].

#### 3.3.3. Social and Context Variables

Young children’s digital experiences and well-being cannot be isolated from their social situations and contexts [50]. First, family SES (income and educational level) was a found to be a significant predictor of young children’s digital use. Surveying 1029 parents of children aged 1 to 9 years, Nikken and Opree found that children’s usage of entertainment media was higher in low-income families than it was in high-income families, and children’s digital proficiency was lower in low-income families than it was in high-income families [81].

Second, at the device level, the content of digital media is also an important factor for young children’s digital usage and well-being. Huber et al. separated 96 children aged 2–3 into three groups and assigned one group to watch an educational television show, one to play with an educational app, and another to watch a cartoon. The results indicated that the type of digital content had a significant effect on young children’s executive functioning performance. The children were more likely to delay gratification after playing an educational app than after viewing a cartoon [82]. Barr et al. found that high exposure to television programs designed for adults during the preschool years was also associated with poorer cognitive outcomes at age 4 [83].

Third, at the policy level, national guidelines regarding digital usage may play a role in young children’s early digital experiences and well-being [50]. In addition, countries differ in terms of their economic development, digital infrastructures, and digital usage policies, which has resulted in the ‘three-level digital divide’ related to access to the Internet (the ‘first-level digital divide’), the promotion of digital skills (the ‘second-level digital divide’), and the specific and tangible outcomes of gaining digital skills (the ‘third-level digital divide’) [84]. Consequently, these three divides can exacerbate social and educational inequalities, which eventually affects young children’s digital well-being.

### 3.4. Early Interventions for Digital Well-Being

The relevant literature regarding early interventions related to digital well-being focuses on “digital well-being” as well as “digital use and well-being”. To explore the scope of existing studies, in this review, we included studies on both of these topics. The existing evidence demonstrates two effective approaches for improving young children’s well-being: (1) digital applications and (2) early interventions.

#### 3.4.1. Digital Applications

More than three hundred apps have been designed to prevent or correct attention deficit and hyperactivity disorder [85]. Some of them have the potential to enhance therapy [86]. Two techniques have attracted significant attention: (1) serious games, referring to technology-based games that enable users to interact, explore, and learn about the world [87], and which are used to obtain more than ‘entertainment’; and (2) exergames, referring to programs that promote healthy behaviors by combining video game technologies and exercise [88]. For example, Play Attention is a computer attention-training system that can measure brain activity and provide feedback in a game-like environment. Existing evidence has demonstrated its usefulness in improving students’ attention, hyperactivity, and executive functioning through one-hour weekly sessions with practice [89]. In exergames, more gross motor activities are required, and the player’s motivation to undertake physical activity increases because of their willingness to succeed in the game [90]. Therefore, players may engage in more physical activities when using digital technologies. These exergames have proved to be effective in increasing physical activity and cognitive functioning and in combatting depression in individuals [91].

#### 3.4.2. Early Intervention Programs

Some studies have reported early interventions aiming to reduce screen time and improve well-being. Schmidt systematically reviewed intervention strategies and identified four school-based and two family-based early education interventions for young children [92]. The four school-based interventional studies were all randomized controlled trials conducted in the United States, and three tested the effectiveness of the “Hip Hop to Health” intervention program, which aimed to prevent obesity among black preschool children through a teacher-delivered intervention. The results indicated that through the “Hip Hop to Health” intervention program, the total screen time was reduced to 28 min per day [92]. The two family-based interventional studies were also conducted in the United States. One [93] successfully reduced the screen time of young children (4–7 years old) to 17.5 h per week. In contrast, the other study [94] failed to reduce the media usage time (including TV and computer viewing) of children aged 2–5 years old, demonstrating the inefficacy of delivering weekly newsletters and booklets to parents containing information about preschoolers’ feeding practices and physical activity.

However, most of these interventions were conducted in Western countries and were specifically targeted at reducing screen time, especially TV viewing time [95]. In Jones’ systematic review and meta-analysis conducted in 2021, 46 intervention studies were found related to reducing young children’s (under 5 years old) screen time. In these studies, 43% were school-based, 26% were home-based, and only 17% were delivered by teachers. Unfortunately, a meta-analysis of these studies revealed that the effect size was insignificant (SDM = 0.096, 95% CI, −0.00 to 0.20), indicating that these interventions might not have been as effective as expected [95].

## 4. Discussion

The digital lifestyle of a child affects their physical health, academic performance, and emotional well-being; thus, digital well-being is critical to child development [27]. As a path-finding study, in this scoping review we identified the various definitions, measurement tools, contributors, and interventions reported in the existing literature related to this topic. In this section, we discuss these findings.

### 4.1. A Proposed Model of Young Children’s Digital Well-Being

The synthesis of the existing definitions demonstrates that the nature of digital well-being is related to the concept of ‘balance’, and it has three compontents: physical, phycological, and social-emotional well-being. Existing evidence has also demonstrated overlaps between digital well-being, digital literacy, and digital citizenship. However, it is not appropriate to confuse the three concepts. Although, in some studies, these terms have been treated as though they have similar meanings, the core concepts behind these three terms are very different (Table 1). Digital literacy refers to the “knowledge, skills, and attitudes that allow children to flourish and thrive in an increasingly global digital world, being both safe and empowered, in ways that are appropriate to their age and local cultures and contexts” [96]. The key term is ‘competence’, including knowledge, skills, and attitudes, highlighting the various capabilities that people should master in the digital world [27]. Digital citizenship can be described as an individual’s ability to “use digital tools to create, consume, communicate and engage positively and responsibly with others” [97]. Thus, the core of digital citizenship is ‘responsible’ participation in the digital world, underscoring the individual’s behavioral style when interacting with others in the digital world [98]. The realization of digital well-being requires the development of digital literacy, and digital citizenship is the ultimate ideal state of development in regard to digital literacy. In other words, digital well-being and digital literacy concern individuals, whereas digital citizenship refers to the attributes of ‘citizens’.

Most of the studies included in this review defined digital well-being generally, and few were specific to young children, with the exception of Johnston, who tried to explore young children’s digital well-being with a focus on digital play. Therefore, based on the literature review, we have proposed a model of young children’s digital well-being (Figure 2). First, on the personal level [47], the five levels on the horizontal axis display the degree of young children’s digital usage: digital addiction, digital overuse, digital well-being, digital literacy, and digital citizenship. Digital addiction and digital overuse are negative digital experiences that may lead to negative emotions and may challenge children’s development, whereas digital literacy and digital citizenship, in contrast, are positive digital experiences that may facilitate young children’s development and lead to positive emotions. Second, at the social level [47], considering that young children always use digital technology under the monitoring of their parents’ [77], their digital usage is mostly influenced by their parents [74]. Therefore, the vertical axis displays seven levels of parent mediation regarding young children’s digital use: setting rules, supervising usage, restricting usage, the co-usage of technology, actively discussing technology, designing developmental digital activities, and supporting children’s self-regulation. Although, setting rules, supervising usage, restricting usage, the co-usage of technology, and actively discussing technology have been reported in the existing literature [77,78], designing digital activities and supporting children’s self-regulation are strategies that have been newly added in this study. This is because the evidence shows that developmental digital activities might be a useful method to improve children’s digital literacy [99] and self-regulation could be an important skill in regard to children’s digital well-being [100,101]. However, few studies have contributed to the development of knowledge about the usefulness of these two strategies in relation to young children’s digital usage; thus, future studies are needed on these topics. Third, the development of young children’s digital well-being is a dynamic process that changes with mediation by parents, as demonstrated by the curved line in the diagram. Negative mediating strategies have the effect of contributing to young children’s digital addiction and digital overuse (indicated by a gray square). In contrast, positive mediating strategies affect young children’s digital literacy and facilitate the development of digital citizenship (the yellow square).

This proposed model was based on the definition stating that digital well-being is the balanced use of digital technology by young children under the guidance, mediation, and support of important adults (e.g., parents or primary caregivers). This concept of a balanced experience means that children can use digital technology to benefit their development (for accessing information, communication, creativity, and entertainment) and to avoid potential risks (content risks, contact risks, consumption risks, and cross-risks). For young children, the guidance, mediation, and support of important adults are critical. In particular, ‘guidance’ means that important adults can guide the development of young children’s digital literacy, which is not limited to restricting young children’s digital usage. ‘Mediation’ means that important adults can regulate children’s digital use through formulating rules, supervision, imposing restrictions, active discussion, and co-usage. ‘Support’ means that important adults can provide high-quality digital content, design developmental digital activities, and support self-regulation by children.

### 4.2. The Effective Measurements of Digital Well-Being

The lockdowns associated with COVID-19 have dramatically increased young children’s screen time [1,102]; hence, stakeholders (parents, teachers, governments, and so on) are concerned about their digital health. However, our synthesis of the findings presented in the literature implies that there is no reliable scale available to measure digital well-being, especially among young children. The Baby Pediatric Symptom Checklist (BPSC), the Preschool Pediatric Symptom Checklist (PPSC), and the Strengths and Difficulties Questionnaire—Parent Version (SDQ) have been used to measure traditional well-being in several existing studies. In addition, the Problem Media Use Measurement (PMUM) tool developed by Domoff in 2019 has been used to measure digital usage [66]. However, there is a research gap in relation to measuring young children’s digital well-being. First, traditional, offline well-being measurements have not been validated for the digital world [52]. Second, the existing measures of digital experience might not be appropriate in evaluating individuals ‘digital well-being’ because they focus on digital usage [52]. Third, the lack of reliable measurement tools might limit the development of research, policies, and practices. Hence, a tool for the specific and reliable measurement of early digital well-being is needed for future research and practice.

### 4.3. The Contributors to Digital Well-Being

This review revealed that child, parent, and social-context variables contributed to young children’s digital well-being. First, a child’s age, circumstances, maturity level, and social context can contribute to their digital well-being [103,104]. For example, children who are vulnerable offline are more likely to be vulnerable in the digital environment [105], so it is difficult to establish clear causality, as those who already suffer from depression or anxiety may be more prone to digital overdependence. However, there is a significant mismatch between the public discourse and the evidence available regarding the effects of digital usage on children’s well-being [105,106].

Second, parents play a crucial role in young children’s digital usage. The outcomes of early digital usage differ according to the parents’ demographic characteristics, income, and educational level, as well as parental digital usage, their perception and mediation of young children’s digital usage, and so on. For example, Ma et al. (2022) found that parental engagement positively predicted children’s social competence, whereas children’s screen time negatively predicted their social competence [107]. Therefore, parents need to be trained to scaffold and support their young children, to build independence, agency, and empowerment by actively talking about engagement with digital technology [25] so that young children can be given enough knowledge about how to balance their digital usage and form healthy habits to improve their digital well-being [54,108].

Third, social context and content influence young children’s digital use and well-being. The child’s family SES, their country’s economic development, and the level of infrastructure available to them play key roles in regard to young children’s digital access, digital skill development, and the outcomes of their digital usage. Although nearly 40 countries have developed digital use guidelines or policies [109], half of the world’s population is still without digital access (primarily living in developing countries) [110]. Moreover, digital content is an important contributor to young children’s learning, development, and well-being [82,83]. Governments need to strengthen digital infrastructure, supervise and regulate digital media content, and develop policies to ensure equality in young children’s digital usage [111].

### 4.4. Early Interventions and Improvements

In this scoping review, we identified two approaches to reducing young children’s digital usage time and avoiding problematic digital usage. However, the effectiveness of these approaches has not been demonstrated in a scientifically sound manner [92], and some of these approaches have been proven to be ineffective for improving digital well-being [94,95]. Several reasons may account for these mixed results. First, the existing interventions have focused on reducing screen time, which is just one factor influencing digital well-being. Other factors, such as the quality of digital media content and parents’ perceptions and mediating actions, may play more critical roles in young children’s digital well-being [112]. Furthermore, digital usage is no longer limited to screens, as there are many screenless digital technologies (e.g., voice robots, VR, and AI) [33,113]. Second, the existing intervention studies only followed young children for short-term periods, ranging between 2 and 20 weeks, and few studies exceeded six months [92,95]. Therefore, an integrated, long-term, systematic digital literacy improvement program is needed in order to enhance young children’s digital well-being [114].

Furthermore, some new techniques such as human–computer interaction (HCI) and child–computer interaction (CCI), are evolving drastically with the potential to improve young children’s digital well-being [115]. For example, Tilli, which is funded by UNICEF, is a game-based, AI-powered social-emotional learning tool that teaches children the skills needed to stay safe and healthy. It uses machine learning and behavioral change frameworks to measure, analyze, and improve children’s social-emotional well-being [116]. Thus, some new approaches and interventions related to young children’s digital well-being are emerging, and future studies should be carried out to keep track of these new techniques and evaluate their effectiveness.

## 5. Limitations and Implications

Although it contributes to the available knowledge on young children’s digital well-being, this study has several limitations. First, taking into account the information load of this scoping review, in this study we did not review studies on digital overuse or digital addiction, which are also important to consider in relation to young children’s digital well-being. Second, in this study we only reviewed articles written in English and did not explore articles published in other languages, which may have narrowed the scope of this study. Third, regarding our proposed model of young children’s digital well-being, more empirical studies are required in order to confirm its effectiveness. However, in this study we synthesized and evaluated the literature on young children’s digital well-being and thus proposed a valuable model which indicates the main difference between young children’s digital well-being and adults’ and adolescents’ digital well-being, namely, that young children need more support from their parents or other important adults in achieving digital well-being. Moreover, in this study we contributed to the theoretical development of this area by elucidating the current confusion regarding the concepts of digital well-being, digital literacy, and digital citizenship, thus enabling a deeper understanding to be reached regarding the different states of digital use.

This scoping review has demonstrated the need for more research and practices to improve young children’s digital well-being. First, in this scoping review, we found no effective measurement tools for assessing young children’s digital well-being. In future studies, researchers should construct new measurement tools based on the model proposed in this study and establish the norms of young children’s digital well-being through the testing of large samples to anchor future empirical research. Second, in this study, we observed the important role of parents in supporting and mediating young children’s digital well-being. Policymakers need to consider this and provide guidance for parents based on empirical research so that parents can play their part in relation to young children’s digital well-being. Third, although we identified some existing digital applications and early interventions, considering their lack of effectiveness, an integrated, long-term, systematic digital literacy improvement program is needed in order to enhance young children’s digital well-being.

## 6. Conclusions

Through a scoping review of 35 articles, in this study we found no consensus regarding the definition of young children’s digital well-being, and effective measurement tools were also found to be lacking. Nevertheless, both child factors (the duration and place of digital use, child demographic characteristics) and parent factors (digital use, parental perception, and mediation) were found to contribute to young children’s well-being, and there were some effective applications and interventions. In this review, we have contributed to the theoretical development process by mapping existing work on young children’s digital well-being, proposing a model, and identifying the research gaps for future studies.

## Figures and Tables

**Figure 1 ijerph-20-03510-f001:**
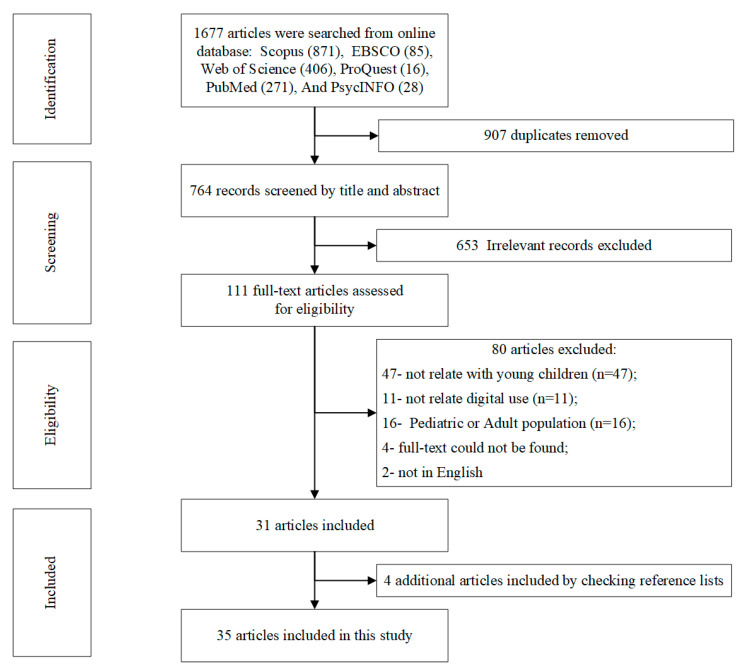
Flowchart of the literature selection process.

**Figure 2 ijerph-20-03510-f002:**
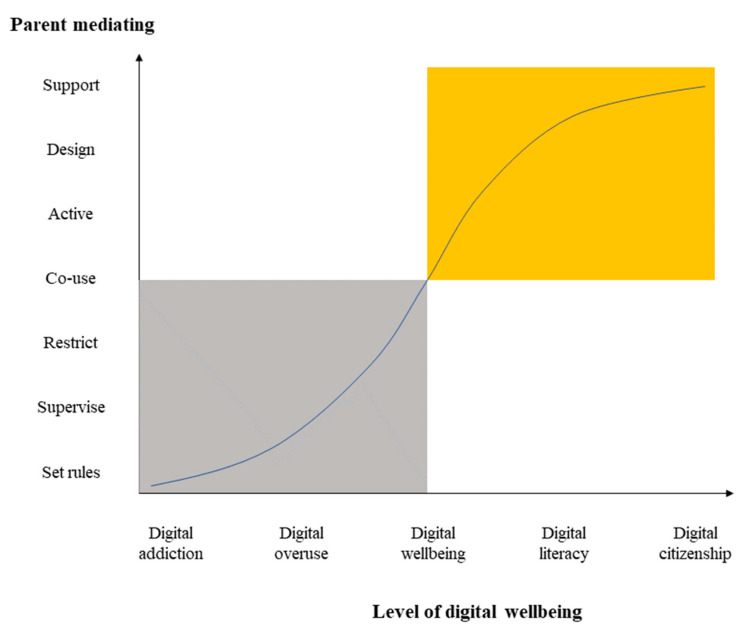
Model of young children’s digital well-being. This model demonstrates the dynamic system of the development of young children’s digital well-being through parental mediation—the curved line across the axis shows the development of young children’s digital well-being. Negative mediating strategies have the effect of contributing to digital addiction and digital overuse among young children (the gray square). In contrast, positive mediating strategies improve young children’s digital literacy and make it possible to develop digital citizenship (the yellow square).

**Table 1 ijerph-20-03510-t001:** Comparison of digital literacy, digital well-being, and digital citizenship.

Concept	Nature	Focus	Level
Digital citizenship	Responsibility	Social and civic participation in the digital world	Behavioral style within society
Digital well-being	Balance	Balance between the adverse effects and benefits of digital usage and experiences	Action of the individual
Digital literacy	Competence	Knowledge, skills, and attitudes that people should master in the digital world	Learning/knowledge of the individual

## Data Availability

Data will be provided upon request.

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
