# Peer review of "A Scoping Review of Digital Well-Being in Early Childhood: Definitions, Measurements, Contributors, and Interventions"

_ijerph, 2023, doi:10.3390/ijerph20043510_

Round 1

Reviewer 1 Report

Dear authors, I congratulate you on the choice of the topic (digital well-being of young children) and on the preparation of the review article. I found its review enjoyable and informative. I do recommend some minor changes, namely

a) remove duplications and shorten

Rationale: Some information from the Introduction (digital well-being of children is not addressed, definition of digital well-being) is repeated in the results.

b) restructure to remove the Discussion section)

Rationale: the content of the Discussion chapter contains the results of your analytical-synthetic work. There does not need to be a Discussion chapter in the review article.

I wish you the best of luck in your future work

Author Response

Point 1: remove duplications and shorten

Rationale: Some information from the Introduction (digital well-being of children is not addressed, definition of digital well-being) is repeated in the results.

Response 1: Thank you very much for this insightful suggestion. The definitions of digital well-being in the introduction part have been integrated into the result part, and the digital well-being of children has been addressed by adding the following sentences on page 2. And the repeated part is also removed from this R1.

“Young children, while increasingly gaining access to digital technologies, are vul-nerable to the associated 'unanticipated risk' that may challenge their maturity [23]”. (line 115-116 on page 2)

Point 2: restructure to remove the Discussion section

Rationale: the content of the Discussion chapter contains the results of your analytical-synthetic work. There does not need to be a Discussion chapter in the review article.

Response 2: Thank you very much for your kind advice. Yes, normally review article does not need a discussion section. However, this article endeavors to develop a new definition of young children’s digital well-being and to identify research gaps. Therefore, we need to have a separate section to do the jobs. Accordingly, in this study, we chose to follow the PRISMA extension for scoping reviews (PRISMA-ScR) checklist to “Summarize the main results (including an overview of concepts, themes, and types of evidence available), link to the review questions and objectives, and consider the relevance to key groups” (Tricco et al., 2018) in the discussion section. Thanks for your kind understanding.

I wish you the best of luck in your future work

Response: Thank you for your professional review and kind advice!

Reviewer 2 Report

This is an important contribution to research on young children's digital technology use, providing a scoping review of studies about digital wellbeing in early childhood. 

The the research design, methods, and positioning of the article is clearly explained. 

The background and definition of digital wellbeing states, that 'the existing literature tends to define the nature of digital wellbeing as ‘a state of balance’ [3], ‘a way of life’ [17], ‘a state of satisfaction’ [18], and ‘a subjective individual experience’ [15].' This summary of definition in existing literature feels underdeveloped, given the references cited include discussion of digital wellbeing not as a state, but as a relationship defined by qualities such as 'resilience' built through education, experience, and support. Please develop this further. to recognise the broader and more nuanced understanding.

Connecting digital wellbeing to concepts and standards of digital literacy and citizenship is appropriate, but could be more clearly differentiated, with the focus more broadly encompassing learning/knowledge, social and civic participation, as well as safety/security, resilience, psycho-social wellbeing.

Reference to APA guidelines is highly relevant, though no mention is made of the once exception these guidelines Mae for young children's digital media use: video calls with relatives etc. this is an important point to acknowledge as it assumes a particular value judgement about what is good or not good internet use for young children. that is, in addition to the dominant rubric of screen time, the more recent amendments privilege human communication over other types of communication or content.

Later in results section, again the definition of DW emphasises balance as key dynamic, (online/offline; benefits/drawbacks; balanced civic way; benefits/risks), but concept of balance feels underdeveloped, because there are different aspects to balance: moderation, context, experience, education; and other balance dichotomies in the literature not noted here: play/learning internet use; active/passive internet use; experience/resilience.

Measurements section reads well.

Contributors section: details both child and parent variables, as well as context such as SE variables. But, missing from here is further acknowledgement and discussion of key contextual variable of access to digital technologies and infrastructure highlighted in research from digital divide socioeconomic literature; as well as further consideration of important variable of content. Media content is important element of much games studies and media studies literature, and needs further consideration here as important contributor to children's learning, wellbeing, and child rights (right to access media, to play etc, have been important points in this UN movement).

The intervention section is interesting addition to the review, though there is not mention of HCI or CCI literature, which directly addresses aspects of designing tech for children.

Overall, this is an important and well-developed analysis, which will make a valuable contribution to the literature once these minor omissions have been addressed.

Author Response

this is an important contribution to research on young children's digital technology use, providing a scoping review of studies about digital well-being in early childhood. 

The research design, methods, and positioning of the article is clearly explained. 

Response: Thanks for your favorable comments!

Point 1: The background and definition of digital wellbeing states, that 'the existing literature tends to define the nature of digital wellbeing as ‘a state of balance’ [3], ‘a way of life’ [17], ‘a state of satisfaction’ [18], and ‘a subjective individual experience’ [15].' This summary of definition in existing literature feels underdeveloped, given the references cited include discussion of digital wellbeing not as a state, but as a relationship defined by qualities such as 'resilience' built through education, experience, and support. Please develop this further. to recognise the broader and more nuanced understanding.

Response 1: Thanks for your insightful suggestion. Yes, the definition should be comprehensive and inclusive. Accordingly, in this R1, we have further developed it to recognize the broader and more nuanced understanding of young children’s digital well-being.  For details, please refer to Line 441-469 on page 6-7.

Point 2: Connecting digital wellbeing to concepts and standards of digital literacy and citizenship is appropriate, but could be more clearly differentiated, with the focus more broadly encompassing learning/knowledge, social and civic participation, as well as safety/security, resilience, psycho-social wellbeing.

Response 2: Thanks for your insightful comment and constructive suggestion. In this R1, we elaborate more on the differences between digital well-being, digital literacy, and digital citizenship. The focus has been shifted to learning/knowledge, social and civic participation. For details, please refer to Lines 915-927 on Page 11, and table 1.

Point 3: Reference to APA guidelines is highly relevant, though no mention is made of the once exception these guidelines Mae for young children's digital media use: video calls with relatives etc. this is an important point to acknowledge as it assumes a particular value judgement about what is good or not good internet use for young children. that is, in addition to the dominant rubric of screen time, the more recent amendments privilege human communication over other types of communication or content.

Response 3: Thanks for your constructive suggestion. We have tried to check the recent amendments of APA guidelines but failed to found it. Could you please share the link?

Point 4: Later in results section, again the definition of DW emphasises balance as key dynamic, (online/offline; benefits/drawbacks; balanced civic way; benefits/risks), but concept of balance feels underdeveloped, because there are different aspects to balance: moderation, context, experience, education; and other balance dichotomies in the literature not noted here: play/learning internet use; active/passive internet use; experience/resilience.

Response 4: Thanks for your critical comments, which are very insightful and helpful. In this R1, we have further developed the concept of ‘balance’. For details, please refer to Lines 479-494 on Page 7.

Measurements section reads well.

Response: Thanks for your kind encouragement!

Point 5: Contributors section: details both child and parent variables, as well as context such as SE variables. But, missing from here is further acknowledgement and discussion of key contextual variable of access to digital technologies and infrastructure highlighted in research from digital divide socioeconomic literature; as well as further consideration of important variable of content. Media content is important element of much games studies and media studies literature, and needs further consideration here as important contributor to children's learning, wellbeing, and child rights (right to access media, to play etc, have been important points in this UN movement).

Response 5: Million thanks for your critical insights and kind advice. Accordingly, in this R1, we have developed four new paragraphs to acknowledge and discuss how the context and content variables influence young children’s digital well-being. For details, please refer to Lines 744-776 on Pages 9-10, Lines 1149-1157 on Pages 13.

Point 6: The intervention section is interesting addition to the review, though there is not mention of HCI or CCI literature, which directly addresses aspects of designing tech for children.

Response 6: Thanks for your kind advice. In this R1, we have added the HCI and CCI in the intervention section and revised the discussion accordingly. For details, please refer to Lines 1218-1225 on Pages 14.

Overall, this is an important and well-developed analysis, which will make a valuable contribution to the literature once these minor omissions have been addressed.

Response: Million thanks for your favorable comments! We really appreciate your professional review and kind help. We hope this R1 is acceptable to you.

Reviewer 3 Report

This is a very important topic. 

It is well written, except for a few language errors. 

First two self-citations in the 'Introduction' section could have been avoided.

A lot of content on definitions in the 'Introduction' section should be shifted to the 'Results' section. 

The focus age group is not clear - is it very young children or less than 18-year-olds?

The discussion section needs to expand to include a more critical discussion. 

Please see the attached document for further comments. 

Author Response

this is a very important topic. 

Response: Thanks for your favorable comments!

Point 1: It is well written, except for a few language errors. 

Response 1: Million thanks for your kind help. We have conducted another round of editing and proofreading. We hope this R1 is free of language errors.

Point 2: First two self-citations in the 'Introduction' section could have been avoided.

Response 2: Thanks for the kind suggestion. Yes, originally, we intended to avoid self-citation. However, the cited work (Dong et al., 2020) has become the most highly cited one on this topic, its citation times is 790 today (See Google Scholar), and it has been listed by the Web of Science as the Global Top 1% Highly Cited Work in 2022. Therefore, it is unavoidable indeed. Nevertheless, in this R1, we have removed one self-citation.

Point 3: A lot of content on definitions in the 'Introduction' section should be shifted to the 'Results' section. 

Response 3: Thanks for your constructive suggestion, which has been fully incorporated in this R1.

Point 4: The focus age group is not clear - is it very young children or less than 18-year-olds?

Response 4: Thanks for your critique. In this R1, we have clearly defined the age range of young children: ages 0 to 8. For details, please refer to Lines 27 on Pages 1.

Point 5: The discussion section needs to expand to include a more critical discussion. 

Response 5: Million thanks for your critical feedback. In this R1, we have expanded the discussion section to make it more in-depth and insightful. For instance, we have elaborated on the HCI and CCI in the intervention section and discussed how the context and content variables influence young children’s digital well-being. For details, please refer to Lines 1149-1157 on Pages 13 and lines 1218-1225 on Pages 14.

Point 6: Please see the attached document for further comments. 

Response 6: Million thanks for your detailed comments, which are very professional and helpful. In this R1, we have addressed the following six comments very carefully. We hope this revision is acceptable to you. Thanks!

  • Introduction section
  • Contributor section: 3.3.1
  • Intervention section 3.4:
  • Intervention section 3.4.2
  • Discussion
  • Discussion 4.1 definition model
